# Searching for the Metabolic Signature of Cancer: A Review from Warburg’s Time to Now

**DOI:** 10.3390/biom12101412

**Published:** 2022-10-02

**Authors:** Pierre Jacquet, Angélique Stéphanou

**Affiliations:** Université Grenoble Alpes, CNRS, UMR 5525, VetAgro Sup, Grenoble INP, TIMC, 38000 Grenoble, France

**Keywords:** metabolic landscape, *metabolic switch*, *metabolic reprogramming*, Warburg effect, reverse Warburg

## Abstract

This review focuses on the evolving understanding that we have of tumor cell metabolism, particularly glycolytic and oxidative metabolism, and traces back its evolution through time. This understanding has developed since the pioneering work of Otto Warburg, but the understanding of tumor cell metabolism continues to be hampered by misinterpretation of his work. This has contributed to the use of the new concepts of *metabolic switch* and *metabolic reprogramming*， that are out of step with reality. The Warburg effect is often considered to be a hallmark of cancer, but is it really? More generally, is there a metabolic signature of cancer? We draw the conclusion that the signature of cancer cannot be reduced to a single factor, but is expressed at the tissue level in terms of the capacity of cells to dynamically explore a vast metabolic landscape in the context of significant environmental heterogeneities.

## 1. Introduction on Metabolism

Cancer cell metabolism is generally considered to be different from normal cell metabolism—the Warburg effect crystallizes this difference in being considered a hallmark of cancer. This review is not about the Warburg effect itself, since there are already many good reviews on this subject. Rather our aim is to consider concepts associated with energy metabolism, limiting the analysis to glycolytic and oxidative metabolism, and to establish how they can be potentially misleading for the understanding of metabolism in the context of cancer.

Metabolism is the set of biochemical reactions that take place in living organisms. We speak of anabolism to designate biosynthetic pathways and of catabolism for degradation pathways. Enzymes are at the heart of these biochemical reactions through catalysis (i.e., the management of reaction kinetics). When several biochemical reactions follow one another in succession, we speak of a *metabolic pathway*. A metabolic pathway can link the reactions that contribute to one or more major functions, but this division remains arbitrary, or at least does not correspond to a real biological separation. Each molecular contributor within a metabolic pathway can also participate in reactions within other pathways. This division nevertheless makes it possible to identify main axes of regulation, to isolate biological functions and to facilitate the exploration of metabolism.

### 1.1. Energy Production Pathways

There are two main metabolic pathways that serve to produce energy in the form of ATP (i.e., glycolysis and oxidative phosphorylation). Glycolysis makes it possible to transform glucose (the main source of energy for the cell), which is transported inside cells, into two molecules of pyruvate in a sequence of ten reactions. The overall glycolysis balance is given by:

Glucose + 2 NAD^+^ + 2 ADP + 2 P*_i_* ⟶ 2 Pyruvate + 2 NADH + 2 H^+^ + **2 ATP** + 2 H_2_O


Glycolysis therefore makes it possible to produce two molecules of ATP from one molecule of glucose. The pyruvate produced can then be reduced to lactate (Figure 1) via lactate dehydrogenase (LDH), but this reaction will not produce more ATP. This sequence of glycolysis—transformation into lactate is commonly called *anaerobic metabolism* because the reactions do not involve oxygen. Lactate is then excreted from the cell by monocarboxylate transporters (MCTs).

Conversely, when pyruvate is decarboxylated to acetyl-CoA (Figure 1) by the pyruvate dehydrogenase (PDH) complex and enters the mitochondria, this is called *aerobic metabolism*. In the mitochondria, it intervenes in the Krebs cycle, generates a molecule of GTP (energetically equivalent to a molecule of ATP) and generates NADH and FADH2. These are then used in oxidation-reduction reactions within the respiratory chain located in the inner mitochondrial membrane. The redox couples NAD+/NADH and FAD/FADH2 release energy which makes it possible to pump protons and create an electrochemical gradient within the inter-membrane space of the mitochondria. This energy potential accumulates until it provides sufficient energy for ATP synthase to transform ADP into ATP. This is the last step in the process called *oxidative phosphorylation* (OXPHOS); it uses oxygen as an electron acceptor. The OXPHOS balance is summarized by:

5 NADH + FADH_2_ + 17 ADP + 17 P*_i_* + 3 O_2_ + 5 H^+^ ⟷ 5 NAD^+^ + FAD + 6 H_2_O + **17 ATP**


From one molecule of glucose, a cell can therefore obtain a theoretical maximum of 38 molecules of ATP. However, experimental estimates show that this number is actually lower (between 30 and 36 molecules of ATP) [1,2]. This is partly due to passive diffusion of protons across the mitochondrial membrane, dissipating the electrochemical gradient that drives the ATP synthase.

### 1.2. Delineations of Metabolic Pathways

Is it always relevant to oppose metabolic pathways to each other based on their functions? As mentioned above, the characterization of metabolic pathways is based on a reductive breakdown. The thousands of reactions that make up cellular metabolism form a network whose complexity can be studied using approaches such as graph theory [3,4]. The metabolic pathways are then viewed as sub-networks (or sub-graphs) of the complete network (Figure 2). In this figure, nodes are metabolites and arrows are reactions. From the original network (leftmost graph), different pathways can be constructed following arbitrary criteria. The middle graph is a two paths construction, the nodes **C** and **E** here are separated from nodes **A**, **B** and **D**. We can imagine further constructions, such as in the third graph, with three pathways, by including or excluding specific nodes.

The problem is to define the best way to delimit these sub-graphs and to justify their existence. Historically, this division was determined by the order in which the reactions of the network were discovered (e.g., glycolysis = glucose transformed into X compounds to finally produce pyruvate). Then a functional division was established (i.e., glycolysis = reactions allowing the production of ATP from glucose to pyruvate). It was already possible to detect a form of arbitrariness in this delimitation. *Why stop at pyruvate? Is glucose the only carbohydrate source?* We can then rely on other arguments, in particular, on the evolutionary structuring of metabolism. *Is this glycolysis block conserved among species? Since when? Are the same molecular contributors involved?* Another approach using graph theory, is to study the robustness of the sub-graph and the conservation of its functions despite observed alterations [5].

Within the scientific community, it is good to agree on the definition of this or that metabolic pathway, although it can lead to arbitrary simplifications (with regard to the biological reality). This is the problem faced by curators of metabolism databases [6]. *How to establish a consensus vis-a-vis publications, new discoveries and definitions established by other databases?* The risk that this division sometimes creates is restricting a metabolite to the function of the metabolic pathway to which it is assigned. Doing so, can conceal its involvement in other reactions. Another danger is to pit metabolic pathways against each other by focusing more on their functions than on the topology of the entire network.

This is exemplified by the aerobic and anaerobic energy pathways described above. Many authors [7,8,9] oppose these two paths. Although the objective is to compare their metabolic fluxes, this creates a dichotomy which implies that the use of one of the two paths excludes that of the other, which is not the case. This dichotomy is frequently employed in the literature associated with cancer.

## 2. The Tumor Singularity

### 2.1. General Characteristics of Cancers

It is difficult to define *cancer* because the properties that characterize it are numerous and vary from one cancer to another. However, common characteristics have been identified among cancers that allow for a deeper understanding of the complexity of this disease. A reference article listing these characteristics is that of Hanahan and Weinberg “The Hallmarks of Cancer” [10,11], published in 2000 and updated in 2011. The authors define ten main axes which are considered as universally present in all cancers (but with varying intensity):independence from proliferation signals,insensitivity to anti-growth signals,escape from programmed cell death,unlimited replication potential,persistent angiogenesis,tissue invasion and metastasis,evasion of the immune system,inflammation,genetic instability,dysregulated metabolism.

With respect to dysregulated metabolism, in the same way that the genome is very heterogeneous within the tumor population, so is the metabolism. The cells experience strong environmental pressures and genetic modifications pushing them to function according to different metabolic modes (in their nature or simply in the distribution of metabolite production flows) from that of the tissue of origin.

Several factors contribute to the progression of cancer in a tumor microenvironment, including the presence of cancer-associated fibroblasts (CAFs), deregulated ECM deposition, chaotic vascularisation, and immune suppression [12,13]. Advanced cancers are characterized by the generation of a hypoxic environment resulting from a lack of correct vascularisation and the activation of a key effector, hypoxia-inducible factor-1 (HIF-1). This factor then promote angiogenesis and vascular remodeling [14]. It is important to note that hypoxia exerts a significant influence on the surrounding cells, as well as the cancerous cells themselves. As a result, certain events in the tumor microenvironment lead to the expansion of aggressive cancer clones [15] and the promotion of lethality.

### 2.2. About Genetic Mutations

It would not be an exaggeration to say that each cancer cell has a unique genetic profile which leads to very high heterogeneity within the tissue. However, there is a prevalence of certain mutations that are frequently found within the same tissue (intra-individual), but also between cancers in different individuals (inter-individual). The best known of the mutated cancer genes is undoubtedly the TP53 gene [16]. In the United States population, this is mutated in 35% of patients (frequency across all cancers) [17]. This gene is a kind of cellular safeguard which, in stressful situations, activates the transcription of a multitude of genes aimed at stopping the cell cycle and repairing any damage to DNA, or, if this is not possible, initiates apoptosis [18,19]. It is then easy to imagine the impact that the mutation of such a gene would have on the appearance or progression of a tumor. PIK3CA (an oncogene) and LRP1B (a gene coding for an LDL receptor) are also mutated in about 15% of cases. The 47 other most mutated genes in the American population are independently present in less than 10% of cases (all cancers combined), and globally in around 5% of cases [17]. In fact, the distribution of each of these genes within different cancer types is very heterogeneous. The TP53 gene is frequently mutated in ovarian (47.8%), colorectal (43.2%) or esophageal (43.1%) cancers, and more seldom in primary leukaemias, sarcomas, testicular cancers or melanomas (5%) [20]. This can lead to two types of reflection: why are these values so low in absolute terms. *Why isn’t a mutated gene found in over 95% of cases for all cancers?* This would make it possible to easily describe a genetic origin of cancer. This is not the case, and the abundant sequencing data show that there is no single common genetic origin for all cancers (at least for the moment nothing points in this direction) [21]. Each cancer will have undergone mutations that disrupt and make a genetic network unstable. A second reflection is the opposite of the first: *Why are the mutation frequencies of some of these genes so high?* Indeed, if we consider that genetic mutations occur randomly, *why are certain genes found more commonly than others and why in such proportions?* There are several answers. The first is statistical: of all the mutations that constantly arise in living organisms, mutations that can disable DNA repair mechanisms are more likely to be observed. Then, the (relative) importance of each gene and its contribution within a cell may depend on its location in the organism. A mutation then does not have a fixed and absolute influence but rather depends on the cellular context. Here again, we would only observe the mutations and their contexts, having induced the appearance of a tumor. Second, tumor mutations are probably not a strictly random process [22]. It is inconceivable to deny the link between cigarette smoking and lung cancer or even between excessive alcohol consumption and oral and pharyngeal cancers [23]. Thus, certain exogenous compounds can have targeted actions on DNA, increasing the frequency of their mutations [24]. Another very important element that has been brought to light in recent years is the infectious origin of many cancers. Mention may be made, for example, of the Epstein–Barr virus as the origin of Burkitt’s lymphoma or cancer of the nasopharynx, the papillomavirus causing cancer of the cervix, or the hepatitis B virus which is implicated in cancer of the liver [25]. The papillomavirus, for example, expresses an E6 protein which targets TP53 to initiate its degradation and an E7 protein which interrupts cell cycle arrest [25].

All these elements make it possible to consider the genetic heterogeneity of cancer, not only with regard to the genetic expression itself, but also with regard to the variety of mechanisms initiating deregulation. The current predominant theoretical approach within the scientific community towards the emergence of cancer, is that cancer is a disease of genetic origin. It is also interesting to quote the excerpt from the conclusion of Hanahan and Weinberg in 2000 in “The Hallmarks of Cancer” [10]:
*“Two decades from now, having fully charted the wiring diagrams of every cellular signaling pathway, it will be possible to lay out the complete “integrated circuit of the cell” upon its current outline. We will then be able to apply the tools of mathematical modeling to explain how specific genetic lesions serve to reprogram this integrated circuit in each of the constituent cell types so as to manifest cancer.”*

Twenty years later, research has effectively refined the map of metabolic networks and attempted multiple modeling approaches in order to test (among other things) the robustness of these networks [5]. However, when we explain the emergence of cancer by the *genetic lesions* that characterize it, we often remain unable to explain why these *genetic lesions* occur. *Is it basically the balloon that breaks the window or the person who shoots into it?* Reading some of the literature one might get the impression that the cause of the broken glass (cancer) has been solved, while only the balloon (the mutation) has been found. After noting that there was not one but several tumor genetic profiles, and, with the need for personalized medicine, the scientific community has understood, over these twenty years, the need to adopt an integrated vision of the cell and to consider the different scales as a whole [26]. This is what the different omic and multi-omic approaches allow, widening the field of research to the transcriptome, the proteome or the metabolome [27]. In general, it is difficult to understand how cancer works without looking at the whole cell, from its DNA to its immediate environment.

### 2.3. The Warburg Effect

Otto Warburg, German physician and biochemist, Nobel prize winner for medicine in 1931, observed in the early 1920s that cancer cells produce much more lactate (fermentation) than healthy cells. What makes the observation special, in addition to the increase in fermentation itself, is the fact that it also seems to occur in the presence of oxygen. Normally, a healthy cell with oxygen available tends to use the pyruvate produced during glycolysis in the mitochondria rather than producing lactate. In 1924, Warburg therefore hypothesized that there must be a defect within the cancer cell preventing it from properly using the mitochondria to produce energy by consuming pyruvate and oxygen [28]. In 1956 he presented his observations and detailed his hypothesis in his article *On the Origin of Cancer Cells* [29]. For him the appearance of cancerous cells was due to this change in metabolic origin. Since then, the hypothesis that mitochondria are deficient, leading to an inability to use oxygen, has been partially invalidated [28]. It may happen that, in some cases, mutations prevent the mitochondria from functioning normally but, in many other cases, the mitochondria is perfectly operational and yet the effect remains [28,30]. There are therefore several emerging questions: *Why do cancer cells ferment so much and why even in the presence of oxygen? Do all the cells of a tumor’s tissue ferment? Do they all have access to oxygen? Are the mechanisms driving cells to ferment within the same tissue all the same?* These questions have been debated in the scientific community for almost 100 years now and have not yielded clear answers, only a few clarifications. Beyond purely technical considerations, the fundamental question facing researchers is: *Is there a metabolic signature that distinguishes a healthy phenotype from a tumor phenotype?* That is what Warburg thought. With the emergence of modern genomics, this question has gradually shifted towards the search for a genomic signature [31]. As we have seen, there is no truly unique genomic signature. Over the past twenty years, the renewed interest in the study of cancer metabolism has spawned thousands of papers on the subject, revisiting Warburg’s discoveries and carrying the hope of segmenting metabolism into healthy and tumor-inducing alternatives, in the same way as was attempted for the genome.

The Warburg effect could confer selective advantages, for example, increased fermentation reducing the activity of the immune system [32] (among other consequences). It is an interesting hypothesis but one that could potentially reverse cause and effect. It is not clear whether the adoption of these metabolic regimes confers any advantage over other cell types or whether it is a collateral effect of the tumor cells inevitably being pushed into specific metabolic modes. In many respects, it could also be said that this metabolic regime confers several disadvantages (increased cell death, for example, through nutrient deprivation) for tumor cells, some of which are still unclear, so it is difficult to conclude whether these should be classified as advantages, disadvantages or side-effects. In the article *“The Warburg Effect: How Does it Benefit Cancer Cells?”* [33], the authors conclude by saying that: “Each of the proposed functions of the Warburg Effect are attractive, but also raise unanswered questions”.

### 2.4. The Concept of Metabolic Switch

A currently dominant idea is that cancer cell metabolism is intrinsically different from that of a healthy cell. It is therefore necessary to identify the associated pathological characteristics. For years, numerous articles [34,35,36] have highlighted the dual nature of cell metabolism: the metabolic term *“switch”*, for example, is used to designate the transition from a metabolism based on oxidative phosphorylation to glycolysis. This *switch* is often used to designate the Warburg effect itself. In this configuration, a healthy cell uses oxygen in the mitochondria and *switches* into fermentation mode by becoming cancerous [37]. It is interesting to note that when the term *metabolic switch* is used to designate the Warburg effect, it is an epistemological error. Indeed, the Warburg effect is the finding of an overproduction of lactate but in no way is it its explanation. The *metabolic switch*, in reality, is the hypothesis that describes a change in metabolic regime, which *de facto* brings out the Warburg effect. The term *metabolic switch* has become so widespread and popular that it has caused semantic and conceptual deviation from what the Warburg effect really is. By mixing observation and working hypothesis, we no longer seek to know if this *metabolic switch* really exists but simply to characterize it. We recall that the main idea expressed by Warburg in his 1956 article [29] was that of the overexpression of fermentation before considering (or not) the presence of oxygen.

Some voices have now raised the issue of the inappropriate use of the term *metabolic switch* [38,39]. This is supported by the recent study by Xiao et al. (2019) [40], who clearly established that cancer cells use both modes of energy production simultaneously, but in different proportions. Hence, there is a broad spectrum of different metabolic states with graded and transient states that do not fit with a switch-like description in which only two mutually exclusive metabolic modes are considered.

### 2.5. Is Warburg’s Phenotype Universal?

The higher glycolytic activity of tumor cells is exploited for cancer diagnosis using FDG-PET (fluorodeoxyglucose positron emission tomography) scanning [41]. Does it means that this higher glycolytic activity is universal in all cancer cells? It is not as simple as that. First, the use of radiopharmaceutical analogues of glucose only ensures detection of a significant tumor mass at the tissue level and not the full detection of all cancer cells in all types of cancer [42,43]. Second, aerobic glycolysis is also found in physiological contexts, such as brain activation [44] or in immune processes [45]. Third, the Warburgian phenotype can be reversed to a non-glycolytic phenotype under lactic acidosis which inhibits the activity of most glycolytic enzymes [34]. Acidity thus acts as a feedback inhibition mechanism. Extracellular lactate can also be used as a substrate by the cells—converted back into pyruvate, it fuels de novo the Krebs cycle (see Figure 3). Hence the Warburg phenotype is not always expressed in tumor cells [38] and cannot be considered as a universal metabolic marker of the cancer cell.

The metabolic heterogeneity within tumor cells of the same tissue impedes the identification of a metabolic profile common to all cells [40,42], and the overall metabolic profile of a tumor tissue does not appear to be fundamentally different to that of their healthy equivalent [46]. Therapies would benefit greatly from the identification of a metabolic target. However it seems that a combination of factors, rather than a single target, would be more probable. The mechanisms responsible for the inverted intra/extracellular pH gradient in tumor cells provide a promising lead [47,48,49,50,51]. These reflections led to the emergence of new questions: *How can tumor metabolic heterogeneity emerge? Does metabolic heterogeneity need to have a genetic origin (mutations) to exist?*

## 3. Differences around the Concepts Associated with the Warburg Effect between Its Discovery and Now

### 3.1. Warburg’s Observations in 1956

In his publication, Otto Warburg described the various experiments he carried out. He showed the increase in fermentation (production of lactic acid) and he measured the amount of ATP produced by cancer cells in rats and mice. He distinguished several behaviors: cells that ferment and manage to maintain the same level of ATP as healthy cells, and cells that do not maintain these levels of ATP and end up dying from *lack of energy*. For him, the scenario for the genesis of cancer cells was as follows: the respiration of healthy cells is damaged (chemically or by successive iteration of induced asphyxia). The cells then find themselves in a state of stress where they no longer breathe properly and try to ferment more and more to compensate for the resulting lack of ATP. During this process of increasing fermentation, occurring very slowly and progressively over several cell generations (several years for humans), many cells die unable to produce enough energy. When the cells finally ferment by producing enough ATP to compensate for the little respiration they have left, they then fully adopt the cancerous phenotype which he described as *dedifferentiated*.

It is striking to note that there is a discrepancy between Otto Warburg’s observations of 1956 and how he is quoted today. Figure 4 illustrates the different keywords associated with the keyword *“Warburg effect”* in articles from 1969 to 2022. The current description taken from the scientific literature, in its broad sense, varies according to the authors, some dwelling more on the production of lactate [52], others on the transition from one type of metabolism to another [53], but generally corresponds to the idea of *deregulation of the metabolism towards aerobic glycolysis*. This definition, in itself, does not appear to pose too many problems because it underlies several premises that are sometimes stated, sometimes not, but which should be clarified (the list is not exhaustive), as follows:Healthy cells primarily use respiration to generate ATP in the presence of oxygen [54,55].Healthy cells make little use of fermentation in the presence of oxygen because ATP is sufficiently produced by respiration [55].If lactate production is high in the absence of oxygen, this is normal since the cell must produce ATP to survive.If lactate production is high and oxygen is available, it is abnormal.Cells favor a particular mode of energy production.The reprogramming of energy metabolism is a dysregulation [56].

The Warburg effect is an observation, that of an increase in the production of lactate (and a greater consumption of glucose) in cancer. Insofar as the mechanisms that lead to this observation are not yet well known, defining the effect by its explanation rather than by its tangible observation is a source of bias. Warburg himself sought to explain his observations by hypothesizing that the mitochondria were dysfunctional, when we know that this is mostly not the case, respiratory activity being conserved in many cell lines [28,30,57,58]. We note, however, that although there have been these findings on mitochondrial integrity, they are still subject to debate within the scientific community, with some papers pointing to a potential mitochondrial origin of the Warburg effect [59,60]. Thus, we note contradictions between the observations of Warburg (which have, of course, been revised since), and what can be read in the literature in texts quoting him, especially regarding the presence of oxygen.


*“I shall not consider aerobic fermentation, which is a result of the interaction of respiration and fermentation, because aerobic fermentation is too labile and too dependent on external conditions. Of importance for the considerations that follow are only the two stable independent metabolic processes, respiration and anaerobic fermentation-respiration, which is measured by the oxygen consumption of cells that are saturated with oxygen, and fermentation, which is measured by the formation of lactic acid in the absence of oxygen.”*


Warburg further insisted on the notion of progression between the two metabolic phases:
*“The mysterious latency period of the production of cancer is, therefore, nothing more than the time in which the fermentation increases after a damaging of the respiration. This time differs in various animals; it is especially long in man and here often amounts to several decades, as can be determined in the cases in which the time of the respiratory damage is known for example, in arsenic cancer and irradiation cancer.*...*Since the increase in fermentation in the development of cancer cells takes place gradually, there must be a transitional phase between normal body cells and fully formed cancer cells.”*

Eventually the notions of *switch* and *reprogramming* (Figure 5) took precedence [57], suggesting genetic modification at a point in time rather than the progressive acquisition of a new phenotype:
*“The Warburg effect is instead a crucial component of the malignant phenotype and a central feature of the ‘selfish’ metabolic reprogramming of cancer cells which is considered a “hallmark of cancer” (Hanahan & Weinberg, 2011). The switch to aerobic glycolysis (i.e., the conversion of glucose to pyruvate) followed by lactate formation is acquired very early in carcinogenesis (oncogenesis), even before tumor cells are exposed to hypoxic conditions”* (Vander Heiden et al. 2009).

The aim here is not to point the finger at all the articles in contradiction to the first observation but to highlight the risk of confusion when the definition includes the beginnings of explanations that change over time. The concept of metabolic switch began to emerge at the beginning of the 1990s (Figure 5) when the genome was more and more perceived to be the key to explaining the cell (as a reference, the Human Genome Project was launched in 1990). The terminology around the Warburg effect evolved from this period. Before, the concepts were little different from those mentioned by Warburg himself. The *concept of metabolic reprogramming* appeared 15 years later.

Figure 6 confirms that research around the Warburg effect during the 2000s focused more on gene expression (increase in the terms *expressions / gene-expression* and decrease in the term *diffusion*, mainly studied in the context of metabolism in the 90s). The term *aerobic glycolysis* came to adopt a very important place in most articles and is the first used in articles today related to the Warburg effect. It might be useful to leave the Warburg effect to its original definition and use new names for the hypotheses and explanations that attempt to explain it. In this sense, speaking of *metabolic switch* and *metabolic reprogramming* is quite correct when their uses aim to clarify a theory (rather than a current of thought) explaining the effect. This makes it possible to discuss, and possibly question, these notions without upsetting the whole lexical field of tumor metabolism.

### 3.2. “Aerobic Glycolysis”

As previously stated, it is widely recognized that *cancer* does not use a single metabolic mode of energy production (ATP) [61]. By *cancer*, however, we can ask what level we are talking about. Cells? Whole tumor tissue? We note that the (technical) capacity to measure metabolic parameters at the cellular scale is central. When articles say for example: *“the cells of our lineage X produce 70% of their ATP by fermentation in the presence of oxygen”*, how can we be sure that each of the thousands/millions of cells in the samples actually produced 70% of their ATP by glycolysis? Or that all cells had good access to oxygen? Had the experimenter measured all these parameters at the cellular level one-by-one? Is this a medium- to large-scale measure? If so, how can we be sure that we are not missing out on an entire metabolically different cellular community? Very often, most measurements are made with devices such as Seahorse XF analysers which measure average extracellular consumption and production of substrates [62], measurements at the cellular scale being very expensive to perform, although this is changing.

It is, thus, interesting to note that, while the accent is placed *on aerobic glycolysis* by insisting on the presence of oxygen for the cells, the literature agrees that tumors are very often hypoxic for a large part of the tissue [63,64]. It is very tempting to think that the cells do not really have access to oxygen and that the tumor tissues are asphyxiated, which leads to an increase in lactate production. Problem solved! The central problem vis-a-vis cancer and more broadly in biology, is the desire to understand the major laws that govern living beings by dealing with the extreme heterogeneity that characterizes them. We must therefore look for unified models in which aerobic and anaerobic fermentation and respiration exist within the same tissue. Cancer functioning in *aerobic glycolysis* is an insufficient model with regard to heterogeneity. This constitutes the first point of questioning of this notion. The second point is related to biochemistry. In itself, *aerobic glycolysis* is nothing abnormal: to feed the Krebs cycle, it is necessary to create pyruvate, which comes mainly from the transformation of glucose through glycolysis. Without it, there would be no (or little) breathing. It would be more accurate to speak of *aerobic fermentation* to integrate glycolysis plus transformation of pyruvate into lactate. This dichotomy between glycolysis and the Krebs-OXPHOS cycle does not exist. It reflects the semantic confusion that surrounds these pathways and encourages the view of two different metabolic modes and, therefore, the idea of *switching* from one to the other. In a genocentric model, the genetic circuit addresses the behavior of the cell. Such a marked change in metabolic fluxes can, therefore, only be the result of a *switch* (in the sense given in electricity: switch/commutator). If such is the case, it is necessary to find how this idea articulates with the very progressive establishment of fermentation described by Warburg. One could argue that time does nothing to prevent a series of mutations or epigenetic changes that lead to this result. However, it seems that this idea is ultimately not compatible with what a switch is, that is to say, the passage from a state A to a state B without any intermediate state between them. This connects with the metaphor of the machine cell.

Towards the end of his career, Warburg said the following (more or less borrowed from Max Planck): *“Science does not progress because scientists change their ideas, but because scientists stuck in erroneous ideas die and are replaced”* [65]. One could smile as to his motives, Warburg being frustrated by the difficulty of getting his ideas, of damaged mitochondria, accepted within the scientific community. One might wonder, however, how much of this statement, if true, is attributable to the way abstract concepts take shape through our words and the expressions we use.

### 3.3. The Reverse Warburg Effect and the Questioning of a Universal Phenotype

The reverse Warburg effect, was a term used by Pavlides et al. in 2009 in the article *The reverse Warburg effect: Aerobic glycolysis in cancer associated fibroblasts and the tumor stroma* [66]. It designates a two-compartment metabolic model in which cancer cells are in a symbiotic relationship with CAFs (cancer-associated fibroblasts). Here, it is not the cancer cells that express a Warburg effect but the surrounding fibroblasts. Cancer cells create oxidative stress that induces this behavior in CAFs (Figure 7). The latter, in turn, excrete lactate captured by the cancer cells which retransform it into pyruvate to replenish the production of ATP in the mitochondria. The hypothesis has gradually been extended by some authors [67,68], integrating a simpler dichotomy: *oxidative cancer cells* and *glycolytic cancer cells*. More broadly, it is a model that describes how the same tumor cancer cells with different metabolic profiles organize themselves to distribute their resources, the production of some being used for the consumption of others.

This two-compartment model is interesting because it suggests that, within this apparent metabolic disorganization, cellular selfishness would give way to a form of symbiotic organization. This organization could be put in place depending on the differences in exposure to nutrients and pH. This need for an alternative model stems essentially from the need to explain the heterogeneity mentioned above. As a result, the Warburg effect is no longer a universal characteristic of tumor metabolism but only one of these expressions. The expression *“metabolic switch”* began to be progressively replaced by the more permissive expression *“metabolic reprogramming”* in the interpretation of a change in metabolic phenotype (Figure 5). We previously dedicated a whole paper to this subject [69].

Thus, rather than insisting on the idea that the cell passes universally from a phenotype A to a phenotype B, the cancerous cell *reprograms* itself towards a phenotype considered abnormal for reasons that we strive to elucidate. The evolution of the concept (which is, however, not yet fully adopted, the idea of switch being still very entrenched) is a step towards taking heterogeneity into account. By forcing the line, we finally end up with two phenotypes (*Warburg* and *reverse Warburg*), which more or less describe the two classic (and physiological) modes of energy production of the cell. Instead of talking about evaporation, we talk about reverse condensation.

The current trend is towards personalized medicine, i.e., finding therapeutic solutions specifically adapted to the characteristics and particularities of each treated patient [70]. It is the correct response to the observation that all tumors are different and therefore must be treated differently. But how different are these tumors? Is there still an interest in trying to understand cancer, if ultimately the variability that constitutes it is greater than its invariability? We can imagine that the problem boils down to the ability to find the nerve center of cancer—the moment in the cell’s existence when, whatever its cause or destination, the cell becomes *cancerous*. If we could determine this precise point, it would be possible to try to avoid it. If this point does not exist, then we would have to resolve to consider cancer as a simple qualifier for similar cellular behaviors. It seems that no consensus in the literature has been found around this hypothetical neuralgic point. There are, therefore, two avenues of study: one which aims to determine the existence of this point by attempting to create unifying models on the emergence of cancer and on the trajectories it can take (the tipping point in time) [71], and the other, which aims to better describe and understand the manifestations of this heterogeneity [72].

## 4. Metabolic Landscape

### 4.1. On the Importance of Heterogeneity

To consider metabolic heterogeneity, it is essential to consider the heterogeneity of the surrounding environment. The microenvironment in which the cells are immersed provides the substrates constituting the entry and exit points of the reactions. In a growing tumor, the substrates that constitute the environment are heterogeneously distributed, from the core of the tumor that experiences starvation, hypoxia and high acidity, to the periphery, where the deprivation of resources is less marked and acidity is reduced. As a consequence, tumor cells do not experience the same metabolic stresses according to their location in the tumor mass and with time. Intercellular interactions through lactate exchange also greatly contribute in the emergence of metabolic heterogeneities [73].

Despite these well-known facts, cell metabolism is often studied in homogeneous 2D cell cultures that ignore spatial heterogeneity. Caution should be exercised in drawing conclusions from models based on an overly homogeneous cell population and global tissue-scale measurements [74]. Measurements made at the scale of the whole cell population only provide an average metabolic picture and hide the spectrum of the different metabolic profiles that can be encountered [75]. This fact was highlighted in a study performed on two cell lines (melanoma and HNSCC), where single-cell and bulk RNA profiles were compared and showed large differences [40]. In this study, OXPHOS variation was the primary contributor to metabolic heterogeneity in malignant and healthy cells and served as a sensor of oxygen availability while stabilizing hypoxia-induced factors (HIF). Beyond spatial heterogeneity, temporal heterogeneity is also paramount, since the environmental conditions constantly change with time. Oxygen variations, for example, are associated with angiogenesis which is a highly dynamic process. It induces cycling hypoxic stresses on the cells of various frequencies and intensities [76,77]. This is far from the binary vision of a switch between two well-defined metabolisms.

In order to fully establish the origin of tumor heterogeneity, it would be necessary to measure and quantify the contributions of genetic-epigenetic factors and of environmental biophysical factors simultaneously. It is expected that the proportions of these cell intrinsic and extrinsic factors would strongly depend on the nature of the tissue and on each individual.

### 4.2. Transient States, Stationary States and Equilibrium States

When we observe a cell, we are often limited by the resolution at which the observations are made. If the concentrations of pyruvate within a cell do not vary, it is necessary to be able to (i) prove that there is indeed no variation, (ii) define whether this absence of variation corresponds to a stationary state, or to an equilibrium. The first point is directly associated with our ability to observe. To characterize a state, we are forced to make at least two observations over time to define the dynamic (derivative), the variation, know the precision of the tools that measure this variation and specify whether this variation, if measured, is significant or not. Regarding the second point, it is necessary to define whether this state is durable over time, at the cellular, tissue or absolute scale. If we know sufficiently well the elements that make up the system, we can deduce its state without needing to perpetuate observations for an infinite time. Let us illustrate this with the following thought experiment: If a bath is emptied at a constant rate and filled at the same rate by a water tank, it is not necessary to wait until the tank is empty to know that the bath will eventually no longer be able to maintain a volume of constant water. However, if it is not known that the bathtub is maintained at a constant volume of water by a reservoir, what must be the characteristic observation time to be taken to define the state of the bathtub?

When we have a dynamic equilibrium, as for the bathtub, we speak of a stationary state, because the apparent invariance of the volume of water results from an equal incoming and outgoing flow. The cell, as in this example, is an open system whose observable reactions are often in quasi-stationary states and depend on the concentrations outside to maintain the concentrations inside. The state of thermodynamic equilibrium does not exist as such in the cell if we look at the complete set of its reactions [78]. Defining a stationary state (dynamic equilibrium) is only possible by specifying a temporal (observed) and structural (reactions, cell compartments, tissues, environment) reference frame. In the bathtub example, the stationary state is defined for a given time (the tank emptying time) and only for the bathtub. If the reference system is now defined as the tank-bathtub set, the stationary state no longer exists. These aspects are important to understand the heterogeneity of a tissue with regard to the cells themselves, as well as their microenvironment. Considering momentarily an imaginary invariability of the metabolism of a cell, of its concentrations kept constant, the microenvironment in which it is immersed is responsible for the history of the cell. The only way to find a stable microenvironment, in a stationary state, is to expand the system even further, by considering the exchanges between the microenvironment and the rest of the organism (typically via the blood vessels). However, what happens if we consider the whole microenvironment-cells as a quasi-closed system? The density of cells, the ambient acidity, the chaotic vascularization of the tissue are elements which, in the context of cancer, create a microenvironment around the tumor which struggles to renew itself and to maintain itself in a stationary state. It, therefore, becomes important to consider the dynamics of the environment as a direct extension of cell metabolism and as a constituent element of the system.

Returning to the Warburg effect, the observation *tumor cells produce a lot of lactate compared to healthy cells*, is too often understood in an absolute way, implying that we are talking about a stationary state of the cells. Is this the case? Looking at the conditions of this statement, we qualify these as follows: *provided that there is enough glucose in the medium or pyruvate in the cells, if there is not too much lactic acid in the medium [34] and if the cells are not of a more oxidative cell type*. It seems that, in view of these considerations, the universal character (in the sense given in the article “The Hallmarks of Cancer” [10]) is not really applicable to the tumor cell. It seems more appropriate to define the Warburg effect as a macro-metabolic or tumor ecology characteristic. It is the dynamics of the tissue and its heterogeneity that brings out this characteristic maintained over time and not at the cellular level. A cell which maintains a high rate of lactate production lowers the pH of its external environment and, thereby, inhibits glycolysis, which produces pyruvate, which generates lactate—if this cell does not return to an oxidative metabolism for whatever reason (lack of ambient oxygen, defective mitochondria), it is inevitably doomed to death. There could be no permanent state in this case.

The focus on the question, *“Why so much lactate in cancer?”*, although legitimate should not make us forget the existence of other states in the tissue. The error that could be made from a theoretical point of view is to grant more importance to the temporality and the prevalence of a metabolic phenotype than to its heterogeneity. It is difficult to experimentally access all the states precisely, especially those existing on shorter time scales, which is why the use of computational models is a real asset. Several models have explored possible states of tumor metabolism and the environment [79,80,81,82,83,84]. A recent model [85], addressed the issue using statistical approaches by attempting to characterize metabolic attractors and their transition zones.

The authors identify the existence of local attractors in this metabolic landscape, which they relate to known phenotypes. They show, in particular, a *proliferative state* associated with a greater excretion of lactate, a *metastatic state* corresponding to a predominantly oxidative metabolism, and a *normal state* which would correspond to a healthy cell phenotype. Each basin of attraction is larger or smaller depending on the epigenetic parameters chosen for the cell and the availability of resources in the environment. In the same way, between each attraction basin, transition corridors are visible. What the authors show by this is that, with the same metabolic circuit, the cell can pass between several metabolic basins (which, moreover, are not limited to well-segmented binary modes), without necessarily having to undergo mutations or complete nutrient deprivation. Their approach is, however, statistical and does not address the temporality of transitions or the spatial distribution of phenotypes within tumor tissue. It is simply the probability for a cell to be in a particular state of the metabolic landscape (or space of metabolic dimensions) according to the conditions to which it is subjected. Genetic-epigenetic modifications will specify the “ease” of passing from one basin of attraction to another.

### 4.3. Homeostasis and Reachability

Homeostasis is defined as a condition or a state in which the living environment (at its different structural scales) is maintained because it is considered optimal and beneficial [86]. We could see this as the set of conditions (pH, temperature, pressure, nutrients, etc.) that allow the organism to maintain itself for as long as possible. Homeostasis with regard to the cell, tissue or organism does not correspond to the search for a perfect and unconditionally stable absolute stationary state, but to an acceptable local minimum for its survival. In the metabolic landscape, this corresponds to maintenance in a basin of attraction. What can a cell do once it is in this state? We can think of two possibilities. The first is to specialize for that state. This would amount to isolating superfluous mechanisms that have no use at the present time and which should not have any in the immediate future. This minimizes the energy needed for cell function and increases cell survival. The second possibility is to temporarily accept internal variability (mutation type) or experience (from the environment) to find an even more stable state. This second point is all the more true at the level of a population. Indeed the only way to get out of an attractor is to accept this variability. The articulation of these two mechanisms, which finally describe the trajectories carried out within the phenotypic landscape, can be achieved through two intrinsic components and one extrinsic component [87]:the robustness of the cellular phenotype: i.e., the conservation of functions despite environmental variations, by multiple redundancies at the level of the genetic code, in the role of amino acids or metabolic pathways and checkpoints,the system evolution: the ability to mutate and authorize new behaviors/functionings of the biological system,the viability: all the environmental constraints of a physico-chemical, radiative, mechanical, social nature, etc. These constraints contribute to test and challenge/experience the phenotype.

From these three components, the notion of reachability emerges, which describes the ability of an organism to evolve towards new behaviors or phenotypes, while maintaining its general functions and ensuring its survival [87]. We can imagine that, in the case of cancer, heterogeneity is the resurgence to the extreme of these components, each being involved in one way or another in the processes involved. Cellular homeostasis may not always be compatible with the homeostasis of tissues or organisms. The balance or imbalance generated by the search for homeostasis at different structural scales of the organism (especially in multicellular organisms) can be a driving force in the exploration of the phenotypic landscape.

## 5. Conclusions

Finally, we could argue that there is no metabolic signature for the cancer cell itself, but that there is a metabolic signature of cancer that only make sense at the tissue level, which requires consideration of the whole tumor environment. What marks the difference from the normal context is the sum of the environmental deregulations, starting from a hypoxic environment which leads—in the normal way—to an increase in glycolysis (cancer cell or not), and to a rise in acidity level. Acidity plays a role in metabolism but this aspect is often underestimated. In particular, it reduces the action of glycolytic enzymes and, therefore, glycolysis, which, added to the increased mortality, can induce a return to the availability and use of oxygen. Within the complex 3D structure of a tumor, the gradients of resources that are established, and that constantly evolve, to accompany the growth of the tumor and the death phases induced by hypoxia and acidity, amplified by therapy, lead to strong dynamically evolving environmental heterogeneities, that are truly the hallmarks of cancer.

At the cell level, a whole landscape of metabolic behaviors are encountered, and, in the process of selection (in the Darwinian sense), more aggressive phenotypes emerge in the long run and enhance the phenotypic metabolic differences compared to normal metabolism. In conclusion, the cancer metabolic signature cannot be reduced to a single phenomenon, the Warburg effect. Instead, the whole dynamic context must be considered and the emphasis placed more on the environment than on the cells, which continue to share the same fundamental metabolic characteristics of normal cells.

Concomitantly, we determined that the concepts of *switch* and of *reprogramming* are often unhelpful since they are a caricature of the reality. Rather than a switch, there is a progressive transition from one dominant metabolic state to another. The two states “glycolysis” and “OXPHOS”, do not really fully exclude one another (except in cases of anoxia or saturation of downstream metabolites), since pyruvate produced through glycolysis is needed to fuel TCA-OXPHOS. With respect to reprogramming, we dedicated a paper to this issue [69], highlighting the fact that this term implied genetic mutations which were not necessarily required. Rather than “reprogramming”, the term “adaptability” is more appropriate to describe the true evolution of the metabolic cell state.

Semantic shortcuts can be useful to highlight a phenomenon, but the danger is forgetting that it was a shortcut in the first place. This can be detrimental to understanding of the phenomenon. In the case of cell metabolism, the shortcuts *“switch”* and *“reprogramming”*, and the concept of the Warburg effect itself, are now currently used in a way that together contributes to producing a segmented picture of cancer metabolism: a discretized and exclusive state markedly different from that of a normal state. This is the opposite of the cell-scale reality where many different states co-exist and evolve with a full spectrum of intermediate states. Cancer metabolism is heterogeneous by nature and explores a vast metabolic landscape. It is the way this landscape is explored dynamically that differs from the normal case.

## Figures and Tables

**Figure 1 biomolecules-12-01412-f001:**
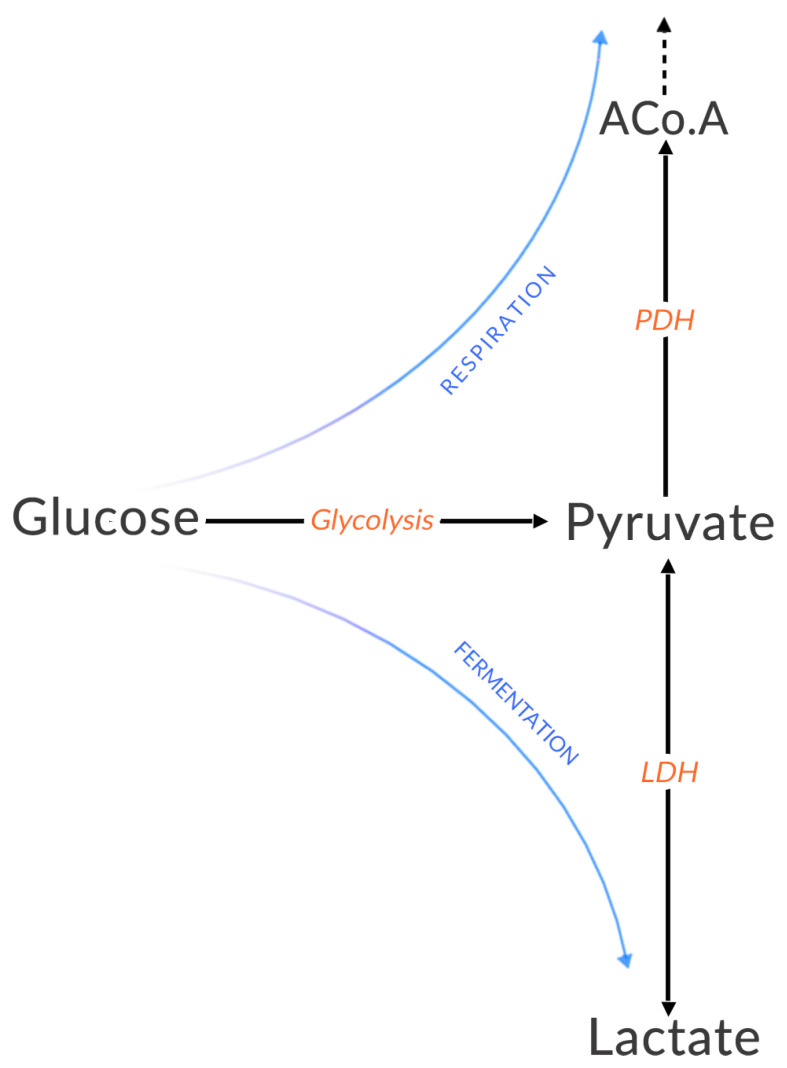
Respiration and fermentation—two processes that begin with glucose and determine the fate of pyruvate. Glycolysis is a sequence of ten reactions that transforms glucose into pyruvate.

**Figure 2 biomolecules-12-01412-f002:**
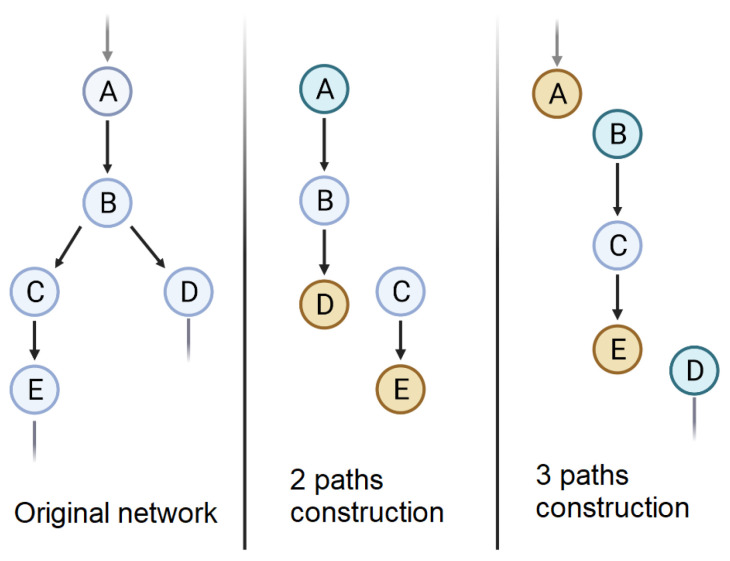
Two examples of metabolic pathways that can be constructed from a simple 5-node metabolic network.

**Figure 3 biomolecules-12-01412-f003:**
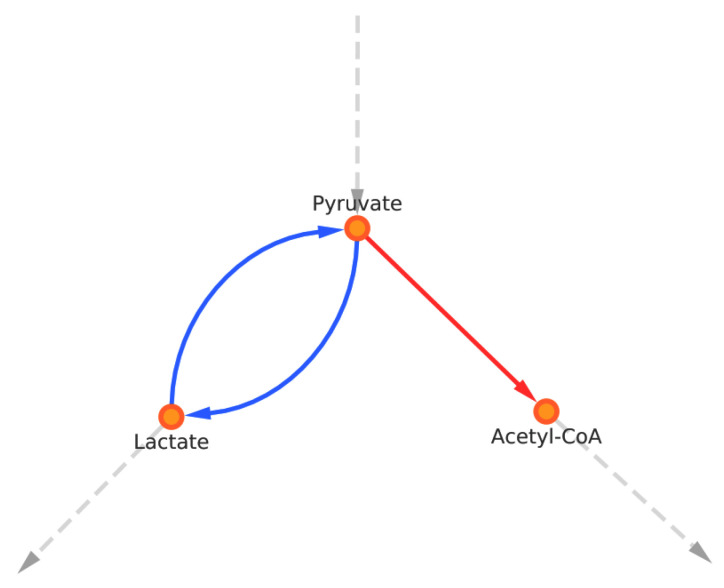
Metabolic branches from pyruvate.

**Figure 4 biomolecules-12-01412-f004:**
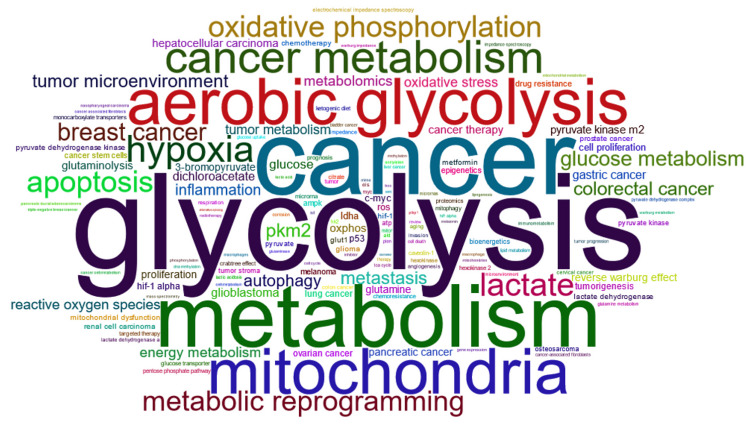
Cloud of keywords provided by the authors of articles dealing with the Warburg effect. This cloud was generated with the R package bibliometrix (version 3.2.1) on 3128 documents extracted from the World of Science database for articles between 1969 and 2022 including the term “Warburg effect” in the title, abstract or in keywords.

**Figure 5 biomolecules-12-01412-f005:**
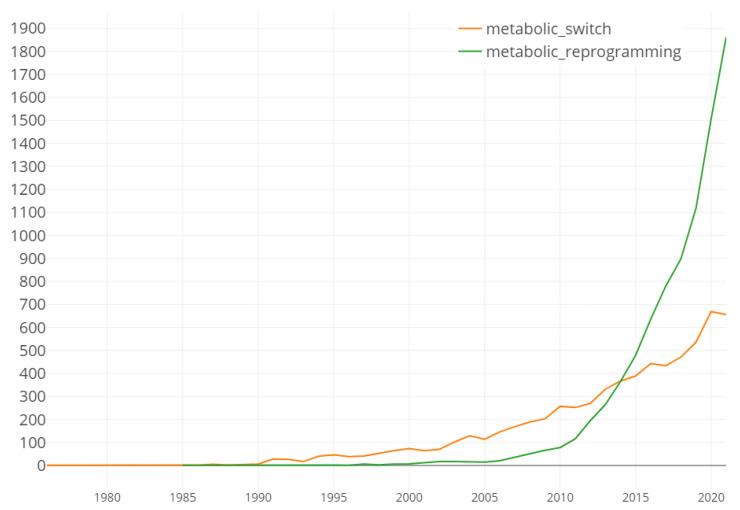
Evolution of the number of publications associated with the terms *metabolic reprogramming* and *metabolic switch*. Results obtained from the World of Science database from their title, the abstract or in keywords.

**Figure 6 biomolecules-12-01412-f006:**
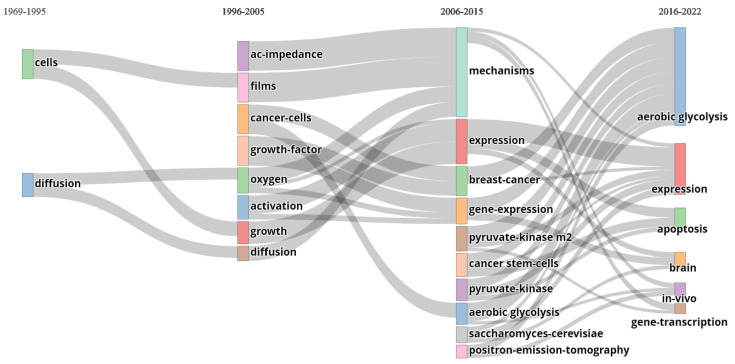
Thematic evolution: keywords on the Warburg effect co-associated within the articles. This Sankey diagram was generated with the R package bibliometrix (version 3.2.1) on 3128 documents extracted from the World of Science database, for articles dating from 1969 to 2022, and including the term *“Warburg effect”* in the title, the abstract or in keywords. Three dates (1995–2005–2015) were chosen to segment the timeline.

**Figure 7 biomolecules-12-01412-f007:**
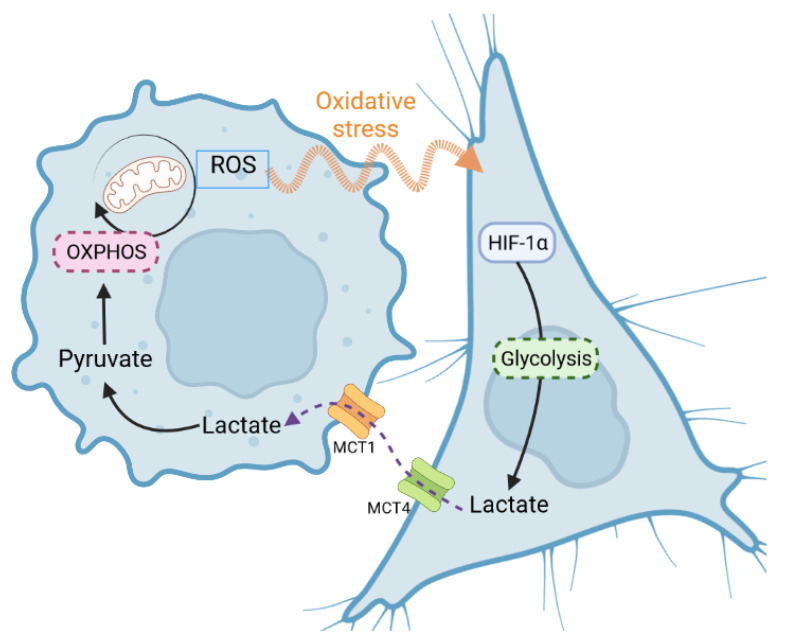
“Reverse Warburg effect”: A cancer cell (**left**) creates oxidative stress in the cancer cell (**right**) by producing ROS (reactive oxygen species). The cell on the right will massively produce lactate which, once excreted, will be used by the cell on the left in the form of pyruvate to replenish the OXPHOS.

## Data Availability

Not applicable.

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
