# Peer review of "Searching for the Metabolic Signature of Cancer: A Review from Warburg’s Time to Now"

_biomolecules, 2022, doi:10.3390/biom12101412_

Round 1

Reviewer 1 Report

In this review, the authors reviewed the evolving understanding of tumor cell metabolism, and concluded that 1) cancer metabolism is heterogeneous by nature and there is no metabolic signature at the cell level but there is at the tissue level; 2) the concepts of metabolic switch and metabolic reprogramming are abusive. This review is interesting. Although I did not agree with the authors in some aspects, I did admit that a deeper understanding of these metabolic terms is necessary. My major concerns for this manuscript are as follows:

1.     The title is “Searching for the metabolic signature of cancer”; however, the authors only introduced the glucose metabolism. How about fatty acids metabolism?

2.     Besides hypoxia, excess fatty acids metabolism may also affect glucose oxidation via the Randle cycle effect. The authors did not consider this.

3.     Page 1, lines 23: “The first, glycolysis…… ”. Which is the second one?

4.     Figure 1: An illustration of glucose metabolism pathways including aerobic glycolysis and glucose oxidation would be better.

5.     Page 2, lines 50: In the part of “Delineations of metabolic pathways and networks ”. What is the meaning of this passage? In addition, Figure 3 is not adequately described, especially the meaning of middle and right panels.

6.     It is difficult to grasp the central idea of the article and the author's understanding of tumor cell metabolism. Highlights of this review should be included in the Abstract or Discussion part

7.     The authors summarized the developing perception of tumor metabolism from Warburg’s time to now. What are the other concepts between Warburg effect (1956) and reverse Warburg effect (2009)? When did metabolic switch or metabolic reprogramming occur? The part “Drift on aerobic glycolysis” is hard to understand and I did not got any evolving conceptions of metabolic terms.

8.      Line 622-624: “The two states glycolysis and OXPHOS never exclude one another (except in the rare case of anoxia) since pyruvate produced through glycolysis is required to fuel OXPHOS. ” I don’t agree with you on this. Excess citrate generated by hyper-activated OXPHOS may inhibit the expression of glycolysis-related enzymes.

9.      Line 465: “hypoxia-induced factors. (HIF). ” An extra full stops mark.

10.  Line 299-301: “Warburg himself sought to explain his observations by hypothesizing that the mitochondria were damaged when we know that this is mostly not the case.” I do believe mitochondria dysfunction promotes the Warburg effect via PDK4. Please give references here.

Reviewer 2 Report

Excellent review by Pierre Jacquet , Angélique Stéphanou “Searching for the metabolic signature of cancer: A review from Warburg’s time to now”. Indeed a great effort to rope in the information about the cancer cell preferences for lactate metabolism despite the presence of oxygen. The authors discussed the cancer cell adaptation to ferment glucose and the metabolic signature of cancer cells

There are a few additional things that need to be addressed:

1>    Section 2.3 :  The role of SIX1 proteins is not discussed in this review.

Genomic Signature (https://www.sciencedirect.com/science/article/pii/S1535610818300102)

A detailed discussion on the involvement of gene machinery in the Warburg effect is missing.

2>    The authors made no comments or descriptions about the role of hypoxia on tumor microenvironment. Also the effect on immune cells infiltrating the tumors. Does it make it advantageous for the tumor to ferment glucose to lactose, maybe to subdue the immune system? Warburg Effect may present an advantage for cell growth in a multicellular environment.

3>    Acidification of the microenvironment and other metabolic crosstalk are intriguing possibilities. Elevated glucose metabolism decreases the pH in the microenvironment due to lactate secretion. What is the author's view on this?

4>    Further, increased glucose consumption is also used as a carbon source for anabolic processes which are needed to support cell proliferation with the excess carbon generated for the de novo generation of nucleotides and lipids. example diversion of glycolytic flux into de novo serine biosynthesis through the enzyme phosphoglycerate dehydrogenase (PHGDH). The review needs to address these angles.

5>    In another study, Bhattacharya et al discuss about the Warburg effect on drug resistance.  

Bhattacharya B, Mohd Omar MF, Soong R. The Warburg effect and drug resistance. Br J Pharmacol. 2016 Mar;173(6):970-9. doi: 10.1111/bph.13422. Epub 2016 Feb 18. PMID: 26750865; PMCID: PMC4793921.

 This seems another great possibility

6>     The role of oncogenes like Myc ( there are several more)  on the Warburg effect needs to be discussed.

7>    The role of hypoxia and the stemness of cancer cells may be discussed. In a way, cancer may get benefited not just metabolically but also to enhance its stem cell pools

Ref : Yun Z, Lin Q. Hypoxia and regulation of cancer cell stemness. Adv Exp Med Biol. 2014;772:41-53. doi: 10.1007/978-1-4614-5915-6_2. PMID: 24272353; PMCID: PMC4043215.

8>    Hypoxia will also promote tumor angiogenenis through HIF pathways. This need to be addressed

9>    Lines 499-520 need to be supported by more references . Most of this appears author's own statement and is not supported by experiments or literature.

10>The review conclusion section must be supported by a graphical model /representation for clarity

Round 2

Reviewer 2 Report

The reviewer is satisfied with the updated manuscript